# Phylogenetic and Timescale Analysis of Barmah Forest Virus as Inferred from Genome Sequence Analysis

**DOI:** 10.3390/v12070732

**Published:** 2020-07-06

**Authors:** Alice Michie, Timo Ernst, I-Ly Joanna Chua, Michael D. A. Lindsay, Peter J. Neville, Jay Nicholson, Andrew Jardine, John S. Mackenzie, David W. Smith, Allison Imrie

**Affiliations:** 1School of Biomedical Sciences, University of Western Australia, Nedlands, WA 6009, Australia; alice.michie@uwa.edu.au (A.M.); timo.ernst@uwa.edu.au (T.E.); 2PathWest Laboratory Medicine Western Australia, Perth, WA 6000, Australia; joanna.chua@health.uwa.edu.au (I-L.J.C.); j.mackenzie@curtin.edu.au (J.S.M.); david.smith@health.wa.gov.au (D.W.S.); 3Environmental Health Hazards, Department of Health, Perth, WA 6000, Australia; michael.lindsay@health.wa.gov.au (M.D.A.L.); peter.neville@health.wa.gov.au (P.J.N.); jay.nicholson@health.wa.gov.au (J.N.); andrew.jardine@health.wa.gov.au (A.J.); 4Faculty of Health Sciences, Curtin University, Bentley WA 6102, Australia; 5School of Chemistry and Molecular Biosciences, University of Queensland, St Lucia 4067, Australia

**Keywords:** Australia, alphavirus, arbovirus, evolutionary analysis, phylogeny

## Abstract

Barmah Forest virus (BFV) is a medically important mosquito-borne alphavirus endemic to Australia. Symptomatic disease can be a major cause of morbidity, associated with fever, rash, and debilitating arthralgia. BFV disease is similar to that caused by Ross River virus (RRV), the other major Australian alphavirus. Currently, just four BFV whole-genome sequences are available with no genome-scale phylogeny in existence to robustly characterise genetic diversity. Thirty novel genome sequences were derived for this study, for a final 34-taxon dataset sampled over a 44 year period. Three distinct BFV genotypes were characterised (G1–3) that have circulated in Australia and Papua New Guinea (PNG). Evidence of spatio-temporal co-circulation of G2 and G3 within regions of Australia was noted, including in the South West region of Western Australia (WA) during the first reported disease outbreaks in the state’s history. Compared with RRV, the BFV population appeared more stable with less frequent emergence of novel lineages. Preliminary in vitro assessment of RRV and BFV replication kinetics found that RRV replicates at a significantly faster rate and to a higher, more persistent titre compared with BFV, perhaps indicating mosquitoes may be infectious with RRV for longer than with BFV. This investigation resolved a greater diversity of BFV, and a greater understanding of the evolutionary dynamics and history was attained.

## 1. Introduction

Barmah Forest virus (BFV), an alphavirus of the *Togaviridae* family, is one of the most medically significant mosquito-borne viruses of Australia [1]. Clinical BFV infection is a serious cause of morbidity, often associated with maculopapular rash, fever, fatigue and debilitating arthralgia or arthritis. BFV disease is near indistinguishable on clinical grounds to the disease caused by Ross River virus (RRV), another medically significant Australian alphavirus [2]. 

BFV was first isolated in 1974 from a pool of *Culex annulirostris* mosquitoes collected in the Barmah State Forest in northern Victoria, and concurrently from mosquitoes trapped in Queensland [3,4]. Human disease was first associated with BFV infection in 1988 [5], and several relatively large-scale outbreaks were identified subsequently in the Northern Territory (1992) [6], Western Australia (1993–94) [7], New South Wales (1994–95) [8] and Victoria (2002) [9]. BFV was thought to be endemic only to Australia, but has recently been sampled from a viraemic child residing in Papua New Guinea (PNG), without an international travel history [10]. BFV cases are reported from every Australian state and it is the cause of approximately 1000 cases per annum, nationwide [1].

The ecological properties of BFV are understudied and poorly understood. It has been long assumed that BFV and RRV share mutual or similar mosquito vector and vertebrate amplifying host species, with limited supporting evidence [11,12]. Often, environmental conditions that support enhanced RRV transmission also bolster BFV activity, but this is not always the case [13,14]. BFV disease epidemics have occurred independent of RRV activity [8,15]. It is speculated that BFV and RRV share marsupials as amplifying hosts, though this notion has not been studied to the same extent for BFV as it has for RRV. RRV and BFV have been isolated from a large number of field-caught mosquito species, approximately 42 and 73 separate species, respectively, suggesting that they are both vector generalists [16,17,18,19]. Important vector species for sustained transmission have not been as thoroughly resolved for BFV as they have been for RRV. 

BFV is the sole member of the BFV alphavirus antigenic complex and is basal to other notable alphaviruses, including RRV and chikungunya virus (CHIKV), on the basis of a midpoint-rooted maximum-likelihood whole-genome phylogeny of all major alphaviruses [20]. Presently there are just four BFV whole-genome sequences publicly available: the 1974 prototype, two contemporary, mosquito-derived isolates from Queensland (sampled 2017 and 2018) and the previously mentioned 2014 clinical isolate from a PNG resident. 

Western Australia (WA) is geographically the largest Australian state, consisting of three broad climatic regions: the temperate south, the tropical/subtropical north and the central arid regions. The majority of WA’s two-million-person population reside in the South West region, where the state’s capital, Perth, is situated. Ecological and surveillance studies, consisting of the trapping, pooling and processing of mosquitoes for the isolation and identification of virus, have been in place in the north and south of WA since 1972 and 1987, respectively [7]. While the population is sparser in the remote localities of WA’s northern regions, disease risk for RRV and BFV is high, particularly in the tropical areas of far-north WA [21]. The procedures involved in the processing of mosquito samples and the identification of virus have been described in detail elsewhere [22]. BFV was first detected in Western Australia in 1989 in the far north of the state, from mosquitoes trapped in Billiluna [7]. The first recognized outbreak of BFV disease in WA occurred in the South West region in the spring/summer of 1993–94 [15]. Over the course of the WA surveillance program, BFV has been isolated over 200 times, but very few viruses have been genetically classified.

A single previous phylogenetic investigation of BFV, based on partial E2 sequence analysis (<500 bp) of 29 isolates sampled 1974–1995, found that despite the geographical and temporal range of sampling, the sequenced isolates were near identical (98–100%) [23]. An avian host was therefore suggested for BFV, given the genetic homogeneity of geographically diverse isolates, indicative of a more mobile host. A robust analysis of BFV evolutionary history and dynamics, through genome-scale phylogeny, has not yet been conducted. A previous temporal analysis of BFV, based on just eight E2 gene sequences, postulated that the PNG isolate diverged from Australian lineages in 1906 [10]. This estimate had a very large margin of statistical error, with a 95% highest probability density (HPD) of 1703–1969. Additional whole-genome sequences are required in order to make a more robust temporal analysis of BFV evolutionary history.

Much work has been done in recent times to expand our understanding of the evolutionary history of clinically significant alphaviruses, through genome-scale phylogeny, to more robustly analyse phylogenetic and temporal relationships of contemporary and historical isolates [24,25,26]. No such analysis has yet been conducted for BFV, despite its significance to the health and economy of Australians. 

In our current study, we conducted a genome-scale phylogenetic analysis of Western Australian mosquito-derived BFV isolates and available published sequences, sampled over a 44 year time period, in order to better understand the evolutionary history and population dynamics of BFV. These findings contribute to our understanding of the transmission dynamics of BFV and illustrate the extent to which these characteristics resemble those of RRV, a virus that is thought to share very similar features. 

## 2. Materials and Methods 

### 2.1. Virus Isolates, RNA Extraction and Sequencing

The WA-derived BFV isolates selected for sequencing are presented in Appendix A. The sampling locations of selected isolates are presented in Figure 1. These were part of a collection of viruses, sourced through an arbovirus surveillance program conducted by the WA Department of Health, in collaboration with the University of Western Australia and PathWest Laboratory Medicine, Western Australia.

For RNA extraction, virus stocks were prepared by a single passage on Vero cell monolayers (ATCC©; CCL-81™) maintained in Dulbecco’s modified Eagle media (Gibco DMEM, Thermo Fisher Scientific, Waltham, Massachusetts, USA), supplemented (by volume) with 5% heat-inactivated fetal bovine serum (FBS), 1% penicillin/streptomycin (Life Technologies, Carlsbad, California, USA) and 1% L-glutamine (Life Technologies). 

Following the observation of cytopathic effect (CPE), 15 mL of Vero cell supernatant was harvested and clarified by ultracentrifugation (4000× *g*, 20 min) in Millipore Amicon Ultra-15 centrifugal units (Merck Millipore, Darmstadt, Germany). RNA was extracted from the concentrate using the High Pure RNA Isolation kit (Roche, Basel, Switzerland) according to the manufacturer’s instructions with some modification. Briefly, 200 µL of concentrate was added to 400 µL of Roche lysis buffer, followed by the standard procedure.

Libraries were generated using the TruSeq stranded mRNA Library Prep kit (Illumina, San Diego, California, USA) according to the provided instructions, altered to exclude steps involving the purification of poly(A)-containing RNA. Superscript III reverse transcriptase (Invitrogen, Waltham Massachusetts, USA) was used to reverse-transcribe the RNA with random hexanucleotide primers (Illumina). Library validation was conducted using the Agilent 1000 DNA kit (Agilent Technologies, Santa Clara, California, USA), before being normalized and pooled. The library pool contained a Phi-X control v3 (Illumina) for a final concentration of 12.5 pM. The denatured pool was loaded onto the MiSeq reagent kit v2, subjected to 300 cycles (Illumina) and sequenced on a MiSeq instrument. 

Reads were demultiplexed and assessed for quality using the FastQC program (V0.11) prior to sequence analysis [27]. CLC Genomics Workbench (QIAGEN, Hilden, Germany) v7.5 and Geneious v11.1 were used to generate full coding consensus sequences, de novo. All contigs over 1000 bp were compared with published sequences available in NCBI using the BLAST function. 

### 2.2. Phylogenetic Analysis

Derived genome sequences were aligned using MAFFT 7.338, available within the Geneious 11.1.4 platform [28]. For phylogenetic analysis, sequences were trimmed manually to remove the highly divergent partial 5′ and 3′ untranslated regions (UTRs), then re-aligned. For all analyses conducted in this study, the four available published BFV whole genome sequences were included. These were strains BH2193 (Accession: U73745, sampled in Victoria 1974), MIDITully (Accession: MK697273, sampled in Queensland 2017), MIDIB78 (Accession: MK697274, sampled in Queensland 2018) and PNG_BFV (Accession: MN115377, sampled in Papua New Guinea 2014). Metadata for these isolates can be found in Appendix A.

Maximum likelihood phylogenies were reconstructed using RAxML 8.2.11 with 1000 bootstrap replicates within Geneious 11.1.4, assuming a GTR+G+I nucleotide substitution model, the most suitable model as determined using JModelTest2 (v2.1) [29,30]. All phylogenies constructed in this study were visualized and illustrated using FigTree 1.4 [31]. Only bootstrap support values of 70% or higher are presented above well-supported nodes. 

### 2.3. Temporal Analysis

Root-to-tip regression analysis was conducted to deduce the temporal structure of our dataset, using TempEst 1.5 [32]. Bayesian Markov chain Monte Carlo (MCMC) analyses were conducted within the BEAST 1.8.4 package in order to estimate the approximate age of the dataset and of any distinct genetic groups characterised [33]. A GTR+G+I nucleotide substitution model was assumed, under a strict clock and a relaxed uncorrelated lognormal (UCLN). The Bayesian skyline coalescent model was used in all analyses, using the default priors. Three independent MCMC chains of 100,000,000 generations were produced and each assessed for convergence (ESS > 200) using Tracer 1.7.1 [34]. Log and the tree files were combined, separately, using LogCombiner 1.8.4 with a 10% burn-in.

Maximum clade credibility (MCC) trees were reconstructed with a 10% burn-in, using TreeAnnotator 1.7.1 [33]. Resultant MCC trees were visualized and illustrated within FigTree 1.4.3, removing posterior probability values of less than 0.7 from poorly supported nodes. Bayesian skyline plots were reconstructed within the Tracer 1.7.1 program with 10% burn-in.

Nucleotide substitution rates and divergence times (time to most recent common ancestor, tMRCA) of major nodes were estimated based on the MCC phylogeny, with statistical error reported as the 95% highest probability density (95% HPD).

### 2.4. Selection Pressure Analysis

The nature and strength of selection pressure acting upon the complete coding region alignment, and individual gene alignments, was evaluated as the ratio of nonsynonymous (dN) to synonymous (dS) mutations (dN/dS), using SNAP v2.1.1 (Synonymous Non-synonymous Analysis program) [35].

Individual sites were assessed for evidence of purifying or diversifying selection pressure using four separate methods available through DataMonkey (www.datamonkey.org/). These were FEL (fixed effect likelihood) [36], FUBAR (fast, unconstrained Bayesian approximation) [37], MEME (mixed effects model of evolution) [38] and SLAC (single-likelihood ancestor counting) [36]. To be considered a site under significant selection pressure required confirmation by at least two methods with the following criteria: *p* < 0.05 (FEL, MEME), *p* < 0.1 (SLAC), posterior probability > 90% (FUBAR). The two separate open reading frames were investigated independently to exclude the 5′, 3′ and junction noncoding regions.

### 2.5. Virus Replication Kinetics In Vitro

The cellular replication kinetics of viruses representing major genetic types were assessed and compared within mammalian (Vero, African Green Monkey kidney epithelium, ATCC © CCL-81™) and mosquito (C6/36, *Aedes albopictus* larvae, ATCC® CRL-1660™) tissue systems.

Virus stock was titrated by plaque assay on Vero cell monolayers for input standardization. For infection, a multiplicity of infection (MOI) of 0.1 was used. Cells were inoculated with the appropriate virus dilution and incubated for one hour (37 °C for Vero cells, 28 °C for C6/36 cells). Inoculum was then removed, and cells were washed vigorously before the addition of the appropriate maintenance media. Development of cytopathic effects (CPE) was monitored and samples collected at regular time points until total CPE was observed in Vero cells, and for a total 84 h for C6/36 cells, which do not typically exhibit CPE. Samples were stored at −80 °C until later analysed. Two independent infections were performed for each virus.

The presence of virus in the timed aliquots was quantified by plaque assay (PA) on Vero cell monolayers. For the PA, each timed aliquot was serially diluted 10-fold. Diluted samples were inoculated onto confluent Vero monolayers in duplicate and incubated for one hour at 37 °C. A methylcellulose overlay (2% FBS DMEM with 1 g/100 mL dissolved methylcellulose) was applied to each cell monolayer, which was then incubated for three days to allow for plaque formation. The virus titre was determined and expressed as PFU/mL (plaque forming units/mL). 

The plaques produced by each virus following infection in the different cell lines were compared for shape, size and general physical characteristics. Plaques were measured from photographs using ImageJ. The diameters of between 30 and 40 plaques produced by each virus were measured at random. 

In this investigation, BFV genotype representatives were compared with representatives of the four RRV genotypes (G1–4), characterised in a recent study [24]. The RRV genotypes characterised in this study are defined by 3.0–5.2% nucleotide sequence divergence [24]. Representative BFV and RRV isolates were all mosquito-derived and had similar, minimal passage history involving alternate passage on C6/36 and Vero cells. All isolates were passaged once on Vero cells prior to use in this investigation.

### 2.6. Statistical Analysis

Virus output values as PFU/mL were log-transformed. The statistical error between the two independent infection experiments is expressed as the standard error of the mean (SEM). Quantitative plaque size results were plotted as boxplots, presenting the mean, the interquartile ranges and the overall range of results for each virus. Statistical significance of viral output and plaque sizes between datasets was determined using an unpaired, two-tailed t-test within GraphPad Prism 8.4.2. A *p*-value of <0.05 was defined as significant.

### 2.7. Recombination Analysis

The RDP4 program (version 4.80) was used to test the BFV whole-genome dataset for evidence of recombination using the RDP [39], Chimeara [40], BootScan [41], 3Seq [42], GENECOV [43], SiScan [44], LARD [45] and MaxChi [46] methods available within the program. Default settings were used for all methods employed.

### 2.8. Accession Numbers

Sequences derived for this study have been submitted to GenBank and were assigned the accession numbers MN689021–MN689047, as shown in Appendix A. 

## 3. Results

### 3.1. Sequence Generation and Gene Region Variability

Thirty novel BFV complete coding region sequences were generated for this study. Most were sequences of Western Australian mosquito-derived isolates, sampled as part of the WA arbovirus surveillance program. Selected isolates represented a breadth of locations within WA, sampled through time (1980–2017: 37 years). Isolates were sourced from eight separate mosquito species. One isolate derived from an Eastern Grey Kangaroo (*Macropus giganteus*), sampled in New South Wales in 2014, was also sequenced. The four available published genome sequences (Accessions: MK697273–4, MN115377 and U73745) were included in the final 34-taxon dataset, sampled over a 44 year time period (1974–2018). A complete list of isolates included in this investigation is provided in Appendix A. 

For all isolates sequenced, only partial 5′ and 3′ UTR sequences were resolved. All sequences were trimmed to remove the highly divergent partial UTRs for all phylogenetic analyses.

The earliest Western Australian BFV isolate in our dataset, KO376-1, isolated from a pool of *Mansonia uniformis* mosquitoes in Kununurra in 1980, was identified at the time as Sindbis virus (SINV; alphavirus, *Togaviridae*) through monoclonal antibody analysis. Whole-genome sequence analysis in the present study revealed this isolate to be BFV, and it is now the earliest known Western Australian isolate of BFV. BFV was previously thought to have been introduced to northern WA in 1989, and subsequently established in the south of the state from 1992. The exact molecular epidemiological history of BFV in WA is now unclear and should be examined further. 

Nucleotide variability was observed in all gene regions of the BFV genome. The average pairwise nucleotide identities and general properties of each of the nine BFV gene regions are presented in Table 1. All regions were equally conserved, with >98% average pairwise nucleotide identity observed in all gene regions across 34 sequences. The *6K* gene was the most variable genomic region between BFV isolates, with an average pairwise nucleotide identity of 98.8%. In a similar analysis of RRV, the nsP3 gene was the most diverse gene region in terms of average pairwise nucleotide and amino acid identity, averaging 95.2% and 96.5% across the 106-taxon dataset, respectively [24]. The nsP3 region of BFV is more conserved in comparison, with an average pairwise nucleotide identity of 99.0% between the 34 sequences. 

The un-gapped BFV genome length, excluding the UTRs, was 10,959 nucleotides, a length considerably shorter than that of similar alphaviruses, including RRV (approximately 11.3kb, excluding UTRs). The gene region sizes of BFV, as compared with RRV and chikungunya virus (CHIKV), is provided in Table 2. The nsP3 gene of RRV and CHIKV is 1.2 times longer in amino acid sequence length compared with the nsP3 of BFV. 

### 3.2. Phylogenetic Analysis

Through maximum likelihood (ML) phylogeny, three distinct genotypes (G1–3) of BFV were characterized (Figure 2). Each major, genotype-defining node of the whole-genome phylogeny (WGS) had high bootstrap values (100% of 1000 bootstrap replicates). The maximum nucleotide divergence between genetic groups was 3.5%, between G1 and G3 isolates. The nucleotide divergence observed between the genetic groups classified in this phylogeny is greater than what was observed between isolates characterised in the only published BFV phylogeny (98–100% average nucleotide identity) that did not characterise any distinct genetic groups. 

The 2014 Papua New Guinea (PNG) isolate, PNG_BFV, sampled in 2014, was the sole member of the G1 genotype and is basal to the Australian G2 and G3 genotypes. The long terminal branch of this genotype suggests long-term circulation, and the accumulation of many nucleotide substitutions before the lineage was sampled. The G2 and G3 genotypes of BFV were each detected within Western Australia during the study period. Based on the ML phylogeny, the two lineages appear to demonstrate clear temporal distinction, with G2 predating G3. During the presumable time period of active circulation of each genotype, isolates were detected in both western and eastern Australia. Among WA isolates, KO376-1, sampled in the far north in 1980, was the sole member of the G2 genotype, with all subsequent WA isolates (sampled 1993–2017), that were sampled statewide, grouping with G3. Based on our sampling, G3 appears to be the contemporary genotype in Australian circulation.

G3 contained two minor subclades, G3A and G3B. All northern WA isolates in our dataset, sampled after 1980, clustered within the G3B sublineage, together with isolates from the south. The entire sublineage was sampled between 2006 and 2017, with the earliest sampled isolates detected in northern WA (2006–2008). During the time of G3B circulation in the north of WA, G3A isolates were detected in the south (2000–2008). The most contemporary isolates (SW105961 and SW105045, both sampled 2017) of our dataset belonged to G3B. The two contemporary Queensland (QLD) isolates, MIDITully and MIDIB78, also clustered within G3B, suggesting these viruses are in active circulation in both the west and east of Australia. G3A comprised isolates exclusively sampled in south WA within an eight-year period (2000–2008). Basal to these two sublineages were four isolates that were sampled during the first recognized BFV outbreak in WA, between 1993 and 1994. These isolates were highly homogenous, perhaps suggesting a clonal expansion during the outbreak. Though these isolates were collected within a narrow timeframe, diversity has been observed in isolates collected within a similar narrow timeframe for the G2A sublineage of RRV [24]. Low diversity may indicate short-term transmission of these variants, having been possibly seeded from another location. 

### 3.3. Analysis of Spatio-Temporal Structure of BFV

To further analyse the geographical diversity and temporal structure of BFV genotypes on a national scale, all geographically classified published full and partial E2 sequences were downloaded for alignment with the 34 BFV taxon database (Appendix A).

The three characterised genotypes of BFV were maintained in this supplemental phylogeny. Isolate PNG_BFV remained the sole member of the G1 group. Both G2 and G3 appeared geographically diverse, containing sequences derived from five and three Australian states and territories, respectively. Both genotypes comprised isolates sampled from throughout Western Australia, as well as from both the east and western regions of Australia. Sublineage G3A remained solely comprised of WA-derived isolates in this revised phylogeny. A contemporary QLD isolate, sampled in 2013, has a basal position within G3, clustering four isolates sampled during the outbreak in the South West region of WA, 20 years prior. 

Interestingly, both G2 and G3 contained contemporary isolates, sampled as recently as 2017 (G2) and 2018 (G3). G2 comprised isolates sampled from 1974 to 1995, plus an additional contemporary sequence, derived from a clinical specimen sampled in QLD (SWBTA40, Accession: MK169388). There was a 12 year period between the sampling of the two most recent G2 isolates, with ample G3 sampling in this interim period. Isolate SWBTA40 was an outlier in root-to-tip regression analysis, and its validity should be considered. G3 overall contained contemporary isolates, sampled 1993–2018, and exclusively from 1996 (with the possible exception of G2 isolate SWBTA40). 

Evidence of spatio-temporal co-circulation of these groups was evident in 1993, with G2 and G3 detection in both the north and south of WA, and again in 1995 with G2 and G3 sampling in New South Wales (NSW). This is the first report of BFV genetic group co-circulation. It is interesting to note that co-circulation was observed within the South West region of WA in 1993—the same year and region as the largest BFV disease outbreaks in state history.

### 3.4. Amino Acid Variability

Unique amino acid transitions were present across all gene regions that could differentiate the three genotypes of BFV (Appendix A). It must be emphasized that G1 is composed of just a single isolate, and so G1-specific amino acid substitutions may be unique to just this isolate. Without more G1 sequences, this is unknown. 

A three-nucleotide insertion, corresponding to a single amino acid, lysine, was observed within the N-terminal domain of the capsid gene of isolate K61404, sampled in Derby in northern WA in 2006. This region of the capsid is highly disordered, containing many positively charged lysine, arginine and proline residues and is involved in RNA–capsid interaction and protein–protein binding. It is interesting to note that K61404 is the only isolate in our dataset derived from *Aedes normanensis*. The addition of another positively charged lysine to the already highly disordered region of the capsid, with association functions, may influence host interactions.

In isolate SW94457, sampled in Harvey (WA) in 2012, a 27-nucleotide (nt) deletion was observed that corresponded to the loss of a hydrophobic nine amino acid sequence: [I/V]GS[V/L][T/P]VGDT. The deletion occurred within the hypervariable domain (HVD) of the C-terminal region of the nsP3 gene, named as such for its lack of conservation among and between alphaviruses. Deletions ranging from 1 to 119 amino acids have been observed in several alphaviruses with minimal effect on the ability of the virus to replicate [47]. Similar deletions ranging from 3 to 135 nt were observed in several isolates of RRV in a recent study, with many (22 of 24 isolates with observed nsP3 indels) resulting in the loss or partial loss of one of the four conserved proline (P*P*PR) motifs within the HVD, a motif involved in host protein binding [24]. The 27 nt deletion of SW94457, described above, occurred 5′ to the first proline motif of the BFV genome. All BFV isolates sequenced in this study contained all four intact proline motifs. The third proline motif in all BFV isolates differed from the classic proline motif configuration, observed in other alphaviruses such as RRV and CHIKV. Rather than the P*P*PR motif, the third residue had either a P*P*PP or P*K*PR configuration. The significance of this alteration is not known but may have influence on the structural configuration of that region.

In all isolates of the G3 genotype, with exceptions, a 25 nt and 58 nt insertion region was observed within the 3′ UTR, 2 and 288 nucleotides from the terminal stop codon, respectively. The earliest isolates that contained these insertions were isolates sampled during the first Western Australian BFV outbreak (1993). The 25 nt insertion was present in all isolates sampled after 1980, except for the QLD isolate MIDITully (2018). Similarly, the 58 nt insertion was present in all G3 isolates, with the exception of the contemporary QLD isolates, MIDITully and MIDIB7 (2017). These, and other notable genome and UTR indels, are listed in Table 3. The 58 nt insertion forms a large hairpin loop, with three internal loops [48]. The terminal loop is rich in poly-A residues. 

All sequences contained read-through opal stop codons within the C-terminal region of the HVD, with the exception of isolate SW31286, sampled in Mandurah in 1993, that contained a tryptophan (W) residue substitution at this site. This is the first observation of such an opal-to-sense codon transition in an isolate of BFV, and the first reported incidence of a naturally occurring opal-to-tryptophan substitution in any alphavirus. Similar opal-to-sense codon substitutions have been observed in RRV, involving the substitution of the opal codon with cysteine and arginine residues [24]. 

While the nsP3 region of BFV contained far fewer nucleotide insertion and deletion events (indels), as compared with RRV, the 3′ UTR was observed to contain many large nucleotide indels, as listed in Table 3. This is in contrast to RRV, which demonstrated fewer and smaller indels in the 3′UTR. Both the 3′ UTR and nsP3 have been implicated in influencing host range.

It is interesting to note that, in comparison with the recently analysed RRV dataset, BFV demonstrated greater variation in the 3′UTR and little variation in the nsP3 region. RRV isolates, in contrast, had only few and short sequence indels in the 3′UTR with many and large indels observed within the nsP3 region. Both the 3′UTR and the nsP3 gene have been associated with influencing host range [49,50].

### 3.5. Temporal Analysis and Population Dynamics

The 34-taxon dataset demonstrated substantial temporal structure (R^2^: 0.92) in root-to-tip regression analysis and so was considered suitable for Bayesian temporal analysis. The WGS/partial E2 dataset did not demonstrate such clock-like behaviour, and so was not analysed in this manner. We reconstructed MCC to estimate tMRCA as well as the evolutionary rates of each of the defined genetic groups.

Given the strong, near linear temporal structure of the dataset, a strict clock model was determined to be suitable for this dataset. Estimates reported hereon are those estimated under a strict clock unless otherwise stated. 

Under the strict model, the root of our dataset and the time of divergence between the PNG and Australian BFV genotypes dates to approximately 1922 (95% HPD 1904–1937) (Figure 3). This early divergence, and the long terminal branch of the G1 node, suggests that BFV has been in circulation in PNG long term and has evolved independently from Australian viruses. The emergence time between the two Australian lineages, G2 and G3, was approximately 23 years. The estimated divergence dates and evolutionary rates of major lineages are provided in Figure 3. Rates estimated under the relaxed clock were similar, with a mean divergence date of 1936 (95% HPD: 1900–1974).

The overall clock rate, assuming a strict clock model, was estimated to be 2.11 × 10^−4^ (95% HPD 1.73–2.52 × 10^−4^). These estimates overlap with those produced under the uncorrelated lognormal (UCLN) molecular clock (mean 2.56 × 10^−4^, 95% HPD 1.57–4.03 × 10^−4^). The rate estimated under the UCLN clock for BFV is approximately 1.3 times lower than the overall clock rate estimated for RRV under the same conditions in a recent study (3.21 × 10^−4^, 95% HPD 2.64–3.77 × 10^−4^) [24]. 

Temporal estimates of BFV variant emergence have recently been presented as part of the PNG BFV isolate investigation [10]. The root of the dataset, comprising eight partial E2 sequences, was dated to 1906, with a wide 95% HPD of 1703–1969. The estimated mean nucleotide substitution rate of the E2 dataset was 1.7 × 10^−4^, with an extremely wide 95% HPD (5.4 × 10^−12^–3.3 × 10^−4^). The estimates produced in our investigation of a 34-taxon whole-genome dataset are more robust in comparison and were comparable with other estimates for alphaviruses [51,52,53].

### 3.6. Population Dynamics

Bayesian skyline plot analysis revealed a near constant population density over time, with little fluctuation in effective population size in plots generated under a strict and UCLN clock (UCLN plot shown in Figure 4). A decline in population size occurred in the early 1990s, then remained stable for two decades before elevating to near-baseline levels in the early 2010s. The relative stability of the BFV population is a stark contrast to RRV, which demonstrated large fluctuations in effective population size, with two major peaks in the early 1980s and 2000s in a similar analysis [24]. Differences in population fluctuations between these two datasets may be an artefact of the difference in taxa contained within each, with more diversity characterised in the larger RRV dataset. 

### 3.7. Selection Pressure Analysis

The mean dN/dS ratio of the entire coding genome of the 34-taxon dataset was determined using SNAP. Across the whole genome, the mean dN/dS ratio was 0.16, suggestive of purifying selection pressure. This is consistent with the widely held “trade-off” hypothesis for arthropod-borne RNA virus evolution; that alternation of replication between two disparate hosts limits the evolution of these viruses, as improved fitness in one host may be detrimental to replication in the alternate host.

Selection pressure analysis of individual sites within both open reading frames, using four distinct methods of selection pressure detection, resolved one site under significant positive selection that was supported by two separate methods. Codon site 297 within nsP1 satisfied both MEME and FUBAR criteria as a site under significant positive selection. This site contained three separate amino acid substitutions: from lysine to glutamic acid (isolate DC57911), asparagine (isolate SW93518) and isoleucine (isolate MIDITully). 

Fewer individual sites were found to be under negative selection pressure throughout the BFV genome as compared with the RRV genome. BFV contained between 9 and 195 sites (all methods) compared with 106–830 resolved for RRV using the same testing conditions [24]. Most sites under negative selection pressure within the BFV genome were detected within the nonstructural polyprotein of BFV.

### 3.8. Recombination Analysis

The alphaviruses are one of the first genera of RNA viruses in which natural recombination was observed [54]. An ancient recombination event between Eastern Equine Encephalitis virus and a Sindbis-like virus gave rise to the Western Equine Encephalitis virus group [55,56]. Recently, Sindbis virus recombinants have been characterised [26]. The BFV dataset was analysed for evidence of recombination to exclude any potential recombinants from phylogenetic analysis. No evidence of recombination was found after using all available methods within RDP4 using default settings. 

### 3.9. Replication Kinetics

Viral growth kinetics of the two Australian BFV genotypes, G2 and G3, were compared through infection of mammalian (Vero) and mosquito (C6/36) cell lines. BFV genotype representatives from Western Australia were selected and matched as closely as possible for passage history. These were isolates KO376-1 (G2, sampled 1980) and SW105961 (G3, sampled 2017). Additionally, BFV growth curves were compared with representatives of the four genotypes of RRV (G1–G4) first characterised by Michie et al. (2020), following infection of the same cell lines under the same conditions [24]. The growth curves of each tested virus following infection with Vero and C6/36 cells are provided in Figure 5.

The most notable contrast in BFV and RRV replicative kinetics was observed in the C6/36 cell line. Each RRV variant replicated at a similar rate in C6/36 cells, but in comparison with BFV, replicated at a much faster rate to reach a significantly higher titre—more than 4-log greater by 24 h post-infection. The maximum RRV titre was reached by 24 h post-infection and was maintained for the duration of the infection time course. In contrast, the viral titre of BFV rose gradually, almost linearly on the log scale, for the duration of the infection. By the end of observation, the titre of BFV still had not exceeded that of any RRV genotype. In the mammalian Vero cell system, RRV and BFV, of all genotypes, replicated at a comparable rate to produce a similar titre. In comparing the replication kinetics of BFV in the Vero and C6/36 cell lines, the rate of virus production was higher following Vero infection, compared with C6/36 infection. However, virus production following C6/36 infection was longer sustained, and by 60 h post-infection, exceeds viral production by Vero cells. In contrast, RRV replication in mosquito cells occurs at a faster rate compared with Vero cell infection, producing higher-magnitude viral titre of longer duration. These results may indicate that mosquitoes infected with RRV remain infectious with higher viral titre that persists for longer, in comparison with BFV-infected mosquitoes.

Viral production was quantified by plaque assay, through inoculation of C6/36 or Vero cell infection supernatant onto Vero cell monolayers. A size comparison of plaques formed in this assay, for both viruses, is provided in Figure 6. 

For all viruses, there was a significant difference in plaque size produced following infection with both cell line infectious supernatants. For RRV, the plaques produced following C6/36 infection were significantly larger than those produced on Vero cells. The one exception was RRV G2, which produced significantly larger plaques following Vero cell infection. The inverse was true for BFV, with both genotypes producing larger plaques following Vero cell infection in comparison with C6/36 infection. This is in keeping with the viral production results, where RRV produced a higher titre at a faster rate in C6/36 cells, while BFV replication was most rapid, initially, in Vero cells. A comparison of plaques produced for each virus genotype (RRV G1–4 and BFV G2–3) with each cell line supernatant is presented in Appendix A. In all cases, there was a significant difference in plaque size produced between all genetic variants studied. 

For all viruses investigated, complete Vero cell cytopathic effect (CPE) was observed by 48 h post-infection. Although alphaviruses are thought to induce minimal pathology in infected mosquitoes, and by extension in mosquito cell lines, cell abnormalities were observed after 48 h in C6/36 cells infected by each of the four RRV genotypes (Appendix A). The BFV strains did not induce such an effect under the same conditions. 

## 4. Discussion

The compilation of a thirty-four taxon near-whole-genome dataset has allowed for the most robust analysis of the evolutionary properties, relationships and dynamics of Barmah Forest virus to date. In contrast to the prior partial genome phylogenetic analysis that characterised a single, highly homogenous genetic class of BFV, our investigation has characterised three distinct genotypes of BFV, G1–G3, circulating in both Australia and Papua New Guinea (PNG). 

Evidence of temporal and geographical co-circulation of the “Australian” G2 and G3 genotypes is described for the first time. G2 and G3 both demonstrated great geographical range, with both genotypes sampled in various Australian states and territories through time. Periods of spatial and temporal co-circulation of G2 and G3 were noted in this analysis. G3 appears to be the contemporary lineage in Australian circulation, but may be co-circulating with G2 in QLD, based on the 2017 isolation of strain SWBTA40. This was the first detection of a G2 isolate since 1993, and its status as a noncontaminant should be considered. 

Comparison of the evolutionary dynamics of BFV to Ross River virus, viruses that supposedly share similar ecological features including reservoir hosts and mosquito vectors, revealed some possible differences in evolutionary dynamics. Timescale analysis of RRV described in a recent study revealed that novel lineages have emerged roughly every decade since the 1950s [24]. Upon emergence, novel RRV lineages demonstrated vast geographical expansion in a relatively short timeframe, and rapidly displaced previously dominating lineages. In contrast, the emergence times of the two Australian BFV lineages occurred roughly 23–25 years apart. BFV genetic groups were not as disparate, based on nucleotide sequence identity, as the four classified genotypes of RRV. It is currently postulated that the rapid emergence of RRV lineages may be driven by semifrequent population bottleneck events, perhaps as a consequence of transmission through a wide variety of mosquito vector species and vertebrate hosts, driving selection for certain variants within the quasispecies [24]. The slower emergence time of novel BFV lineages is perhaps suggestive of a narrower host range, resulting in fewer population bottleneck events and thus a stable population. RRV and BFV therefore may not share exact transmission characteristics. Alternatively, the true diversity of BFV lineages may be ill-resolved in the current analysis, requiring more sequences from a wider geographical span for robust estimations. Certainly, the ecology of BFV needs to be explored further, with particular focus on the possibility of amplification and dispersal of virus by a more mobile, possibly avian host, as postulated in the previous BFV phylogenetic analysis [23].

Most isolates that were included in this analysis were sourced during routine Western Australian mosquito-borne arbovirus surveillance, conducted by the WA Department of Health and the University of Western Australia. Most samples included in this dataset were sourced from the south-west of WA, with only six isolates available from the northern region, and none from the central arid region. In this analysis it was not possible to analyse the geographical movement of BFV through time in any great detail, but this will be pursued in the future with increased sampling from a wider spatio-temporal range, including other Australian locations. This may reveal locations in which viral diversity is generated, and from where virus is seeded into new locations. 

Prior to this study, BFV was believed to have been introduced to Western Australia in 1989 [7]. Sequencing of isolate KO376-1, sampled in 1980 and identified as SINV at that time, revealed this isolate as BFV, and it is now understood to be the earliest known WA isolate. The epidemiology of BFV within WA is now uncertain. In a BFV seroprevalence study in the human and animal populations in the South West region of WA, it was noted that the seroprevalence of anti-BFV antibody in sera collected prior to 1992, the supposed introduction of BFV in to the South West, was null—with the exception of one equivocal rabbit [57]. However, seroprevalence of anti-BFV antibodies in the population sampled after its supposed introduction was very low, just 0.4% across all sources. 

Our more robust whole-genome dataset was used to derive temporal estimates and dated the divergence of PNG and Australian BFV lineages to approximately 1922, with a relatively narrow 95% HPD (1904–1937). The long terminal branch, of the sole isolate of this lineage sampled as recently as 2014, indicates that this lineage has been evolving locally within PNG for some time, independent of Australian isolates. Previous estimates of the divergence of the PNG and Australian lineages were not well supported with a very large 95% HPD (mean 1906, 95% HPD: 1703–1969) [10]. The dataset generated for the present study was far more robust. 

There are several marsupial species that are endemic to PNG, which may have played a role in the long-term persistence of BFV in the region [58]. A migratory bird flyway exists between far-north Queensland and PNG, which may have been involved in the seeding of BFV from one region to another [59]. The role of birds in the maintenance and transmission of BFV has not been explored in any great depth within Australia. Alternatively, BFV may have been introduced through enhanced movement of viraemic people during and after WW1. Following the war, Australia took administrative claim of the region through Treaty of Versailles peace agreements with Germany, enacted through the 1920 New Guinea Act. The timing of WW2, and the movement of large numbers of Australian troops into PNG, falls within the range of relaxed temporal estimates (1900–1972)–this may be a more likely source of the introduction. The direction of BFV movement cannot be concluded and may very well have been introduced to Australia from PNG.

An interesting observation of this investigation was that of the highly variable 3′UTRs among isolates of BFV. The nsP3 gene of these isolates, in contrast with RRV, was far more conserved with only one indel observed among all isolates. Conserved proline and FGDF-like nsP3 motifs were present in all BFV isolates. The difference in variability between RRV and BFV in the nsP3 and 3′UTR perhaps further highlights the differences in transmission dynamics between these two viruses. Variability in the nsP3 region, and specifically the proline motifs which are involved in host-protein interactions, may allow RRV a degree of host plasticity.

The biological implications of genetic diversity of BFV and RRV were investigated in this study. Here, RRV appeared to replicate at a faster rate and to higher titre for a longer period of time in mosquito cells, compared to BFV. This may suggest that RRV infection in mosquitoes is longer-lasting, with mosquitoes having more infectious days in their life cycle, and more opportunity to transmit virus to a new host. This may explain why RRV is seemingly in higher circulation compared to BFV based on mosquito isolation frequency and disease rates. This analysis, limited to an in vitro model of variant representatives, may not reflect in vivo mosquito infection and is not likely to reflect the virus availability within mosquito salivary glands. Within the mosquito host, virus must penetrate multiple tissue barriers to allow for on-going transmission. The difference between BFV and RRV production in this tissue system is intriguing and should be investigated further with live mosquito models. All viruses studied had similar passage histories that did not involve excessive passage in either mammalian or mosquito-derived cell lines. The viral production curves in both tissues were not significantly different between any of the genetic groups studied. In the plaque size analysis, however, there was significant differences observed between all genetic groups of both viruses following infection of both cell-lines. Interestingly the contemporary RRV genotype in Australia-wide circulation, G4, consistently produced the smallest plaques in both the C6/36 and Vero cell-lines. G2, which was the dominant lineage in circulation in Western Australia prior to the emergence of G4 in the mid-1990s, was the only RRV variant studied that produced larger plaques in the Vero cell line than C6/36 plaques. In contrast, the most heavily represented BFV genotype, G3, consistently produced significantly larger plaques than G2 in both infection studies. G1 of BFV, represented by the single clinical isolate from PNG was not available for in vitro investigation, but this will be pursued in the future. The significance of these observations should be further explored and possible associations these phenotypes bare in relation to fitness advantage, investigated. It would be of interest to investigate the role of the observed large-scale 3′UTR nucleotide insertion and deletions on the replication kinetics of BFV.

Future studies will include whole genome sequences of isolates derived from a wider geo-temporal range, in order to support estimates produced in this investigation, and to better understand the movement and evolution of BFV in Australia. More isolates from PNG would be interesting to investigate, to better understand the history and epidemiology of BFV in the region, and between PNG and Australia. Continued BFV genotyping would be beneficial to monitor the emergence of any new lineages. 

## 5. Conclusions

A greater diversity of Barmah Forest virus (BFV) has been resolved through more robust analysis of a 34-taxon complete genome dataset, with three distinct genotypes characterised (G1–3). In contrast to Ross River virus (RRV), another medically significant Australian alphavirus, the BFV population was more stable with emergent lineages arising between long intervals (23–25 years, compared to 10 years for RRV). BFV in Australia and Papua New Guinea have evolved independently over the past 80 years. 

## Figures and Tables

**Figure 1 viruses-12-00732-f001:**
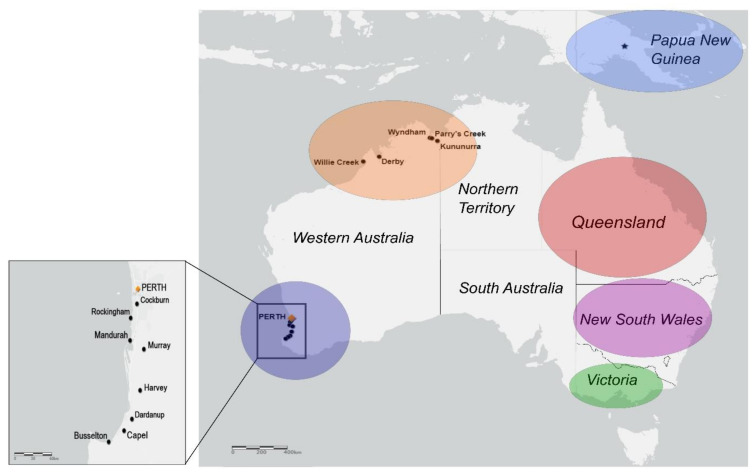
Sampling locations of Barmah Forest virus (BFV) isolates selected for sequencing. Individual locations within Western Australia were known and so are named. Locations were classified into larger, colour-coded geographical regions. Stars indicate locations of clinical isolate sampling, and dots indicate locations of mosquito-derived isolate sampling. Perth, the capital of Western Australia (WA) is indicated by an orange marker. The map was generated using ArcGIS (www.arcgis.com).

**Figure 2 viruses-12-00732-f002:**
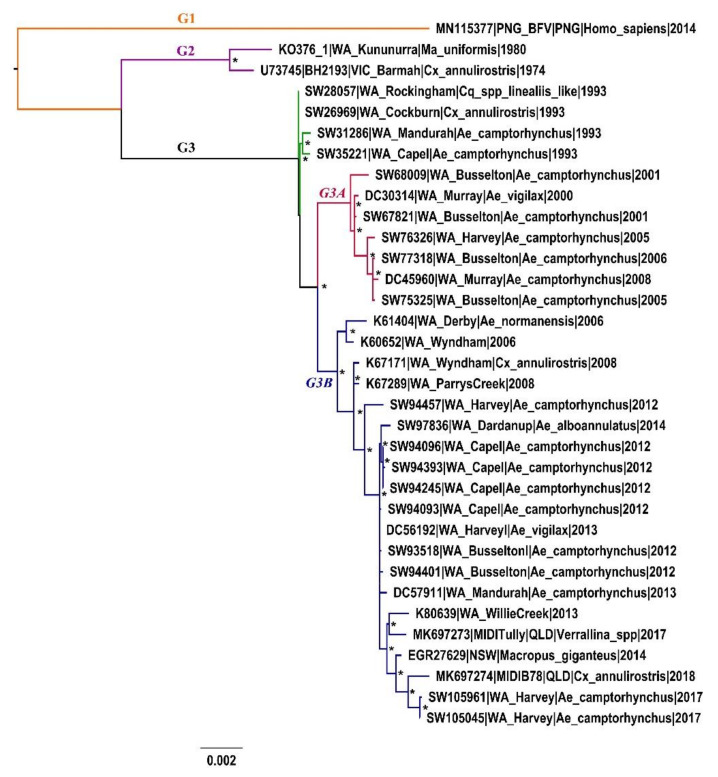
Midpoint-rooted maximum likelihood phylogeny of the 34-taxon complete coding region Barmah Forest virus dataset. The three major genotypes (G1–3) and the minor sublineages of G3 have been labelled. Asterisks indicate nodes with bootstrap values over 70%.

**Figure 3 viruses-12-00732-f003:**
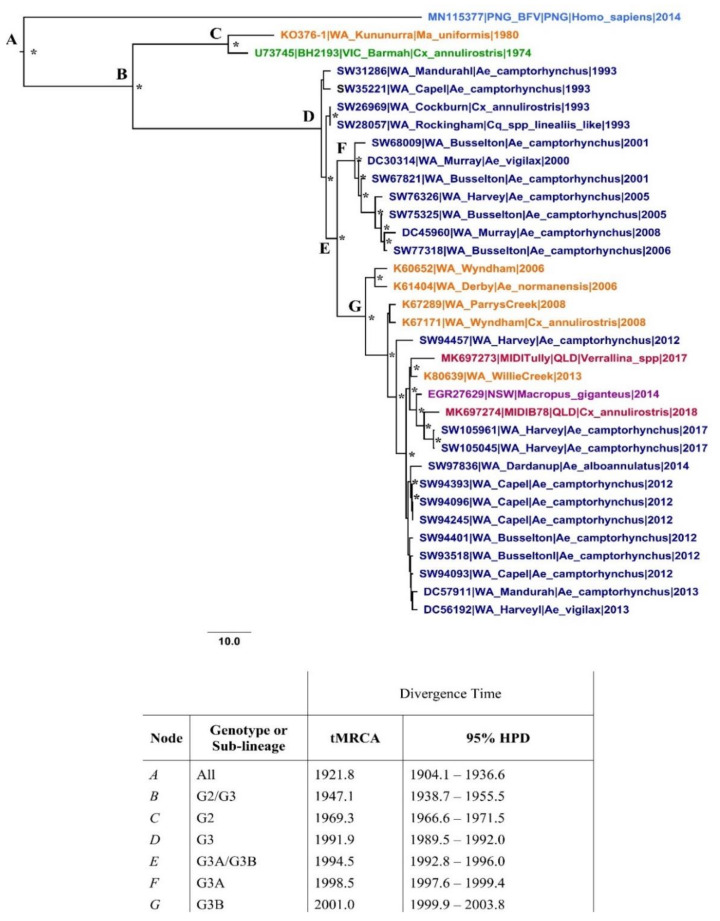
Maximum clade credibility (MCC) tree of the 34-taxon dataset reconstructed assuming a strict molecular clock. Posterior probability values > 0.70 are presented above well-supported nodes (*). Taxa are coloured for their geographical origin, as per Figure 1. The divergence time (time to most recent common ancestor, tMRCA is provided for each major lineage-defining node (A–G). Statistical error is reported as the 95% highest posterior density (95% HPD).

**Figure 4 viruses-12-00732-f004:**
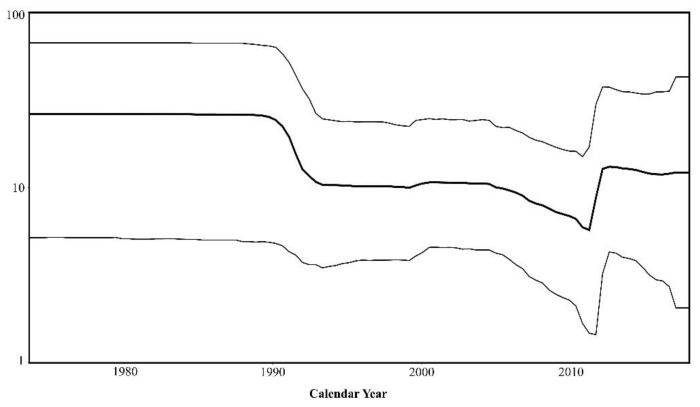
Bayesian skyline plot (BSP) reconstruct assuming an uncorrelated relaxed lognormal (UCLN) molecular clock under a GTR+G+I nucleotide substation model, with default tree priors. Fluctuations in effective population size (y-axis) through time (Calendar years, x-axis) are presented.

**Figure 5 viruses-12-00732-f005:**
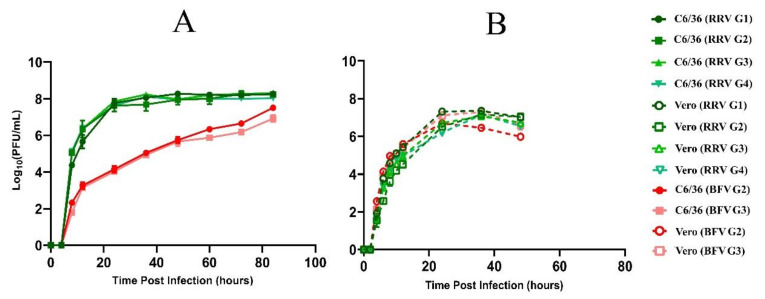
Replication kinetics of representative Ross River virus (RRV, green) and Barmah Forest virus variants (BFV, red) in C6/36 and Vero cell lines, over time. (**A**) Comparison of viral production (Log_10_ PFU/mL) of representatives of RRV genotypes 1–4 (G1–4) and representatives of BFV genotype 2 and 3 (G2–3) in C6/36 cells over time. (**B**) Comparison of viral production (Log_10_ PFU/mL) of RRV G1–4 and BFV G2–3 in Vero cells over time. Bars present the standard error of the mean between two independent infection experiments.

**Figure 6 viruses-12-00732-f006:**
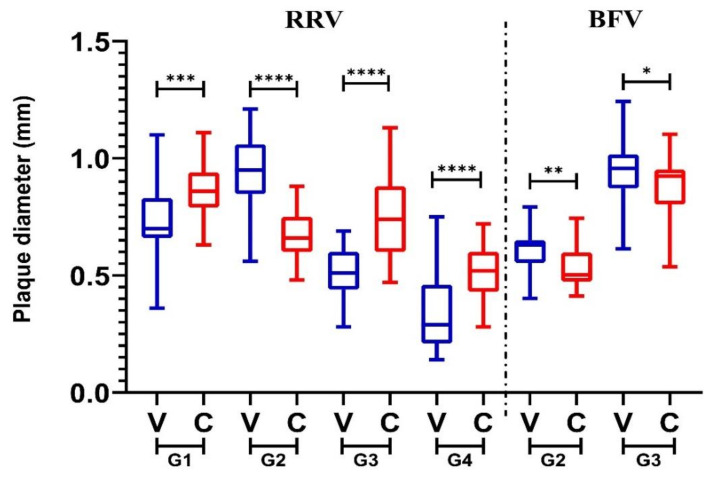
Size (diameter, mm) comparisons of plaques produced following inoculation of Vero cells with C6/36 (C, red) or Vero (V, blue) supernatant following infection with genotypes 1–4 (G1–4) of Ross River virus (left of the line) or G2–3 of Barmah Forest virus (right of the line). Plaque sizes produced in C or V cells were compared for each virus, significance values are shown above these datasets (**** *p* ≤ 0.0001,*** *p* ≤ 0.001 ** *p* ≤ 0.01, * *p* ≤ 0.05) as determined by the student t-test.

**Table 1 viruses-12-00732-t001:** The nucleotide (nt) and amino acid (aa) sequence length (excluding gaps), the average pairwise nt and aa identity (%) and the base frequency of each individual Barmah Forest virus gene region within the 34-taxon dataset.

Gene Region	Gene Region Length Excluding Gaps (nt/aa)	Pairwise Identity (nt/aa) (%)	Base Frequencies, A, C, G, T (%)
nsP1	1599/533	99.6/99.7	29.6, 24.7, 26.3, 19.4
nsP2	2394/798	99.3/99.7	30.2, 25.3, 24.2, 20.4
nsP3	1410/470	99.0/99.3	27.4, 26.1, 25.9, 20.6
nsP4	1833/610	99.2/99.8	31.2, 21.9, 23.2, 23.7
C	762/254	99.4/99.7	32.7, 25.5, 24.0, 17.8
E3	204/68	98.9/98.1	27.0, 29.1, 22.0, 21.9
E2	1263/421	99.4/99.7	28.1, 27.2, 23.1, 21.7
6K	174/58	98.9/99.7	21.5, 22.8, 23.3, 32.3
E1	1320/439	99.6/99.8	27.3, 25.0, 24.1, 23.6

**Table 2 viruses-12-00732-t002:** Comparisons of gene region nucleotide sequence (and amino acid sequence in brackets) lengths between Barmah Forest virus (BFV), Ross River virus (RRV) and Chikungunya virus (CHIKV).

Gene Region	BFV	RRV	CHIKV
nsP1	1599 (533)	1602 (534)	1605 (535)
nsP2	2394 (798)	2394 (798)	2391 (797)
nsP3	1410 (470)	1650 (550)	1581 (527)
nsP4	1833 (610)	1833 (611)	1836 (612)
C	762 (254)	810 (270)	783 (261)
E3	204 (68)	192 (64)	192 (64)
E2	1263 (421)	1266 (422)	1269 (423)
6K	174 (58)	180 (60)	183 (61)
E1	1320 (439)	1314 (438)	1317 (439)

**Table 3 viruses-12-00732-t003:** Nucleotide insertion and deletions (indels) within the nsP3 and 3′UTR of the 34-taxon BFV alignment. The genomic position, indel size and isolates with the reported indel are presented. The location of 3′UTR indels is expressed as the number of nucleotide (nt) residues it is from the terminal stop codon.

Location of Indel (Gene Region)	Size of Indel	Isolates With indel
5050–5076 (nsP3)	27nt deletion	SW94457
7440–7442 (C)	3nt insertion	K61404
3′ UTR		
2nt post-terminal stop codon	25nt insertion	All G3, except MIDITully
40nt post-terminal stop codon	18nt deletion	KO376-1
57nt post-terminal stop codon	5nt deletion	SW77318, SW76323, DC45960, SW75325
115nt post-terminal stop codon	14nt deletion	PNG_BFV
165nt post-terminal stop codon	11nt deletion	EGR27629, SW10545, SW105961, MIDIB78
288nt post-terminal stop codon	58nt insertion	All G3 except MIDITully and MIDIB78

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
