# Peer review of "Phylogenetic and Timescale Analysis of Barmah Forest Virus as Inferred from Genome Sequence Analysis"

_viruses, 2020, doi:10.3390/v12070732_

Round 1

Reviewer 1 Report

In this study by Michie et al., the authors performed a comprehensive whole genome phylogenetic analysis of 30 new Barmah Forest virus (BFV) isolates from samples collected over time across Australia.  The authors performed thorough phylogenetic and sequence analyses on each viral protein and compared these findings to Ross River virus (RRV) and chikungunya virus. Using these data, they identified a number of nucleotide deletion and insertions from different BFV isolates that they speculate could have a role in the viral life cycle. Finally, the authors address the replication kinetics of representative strains of BFV and RRV in Vero and C6/36 cells to understand if there were any growth differences between these viruses. Interestingly, they found that BFV grew slower on C6/36 cells compared to RRV. The study is well-written and well-executed. It provides novel insight into BFV diversity and evolution and offers interesting ideas in viral factors that could impact BFV biology. However, there are several issues that I feel could strengthen the manuscript.

1. Section 2.1 – I think this section would fit better in the introduction as it provides valuable facts about Australia and BFV and does not provide an "method"

2. I know it is referenced, but the authors should include details about RRV in the materials and methods. Are these viruses from an infectious clone, an isolate, how were they generated (passaged on Veros as well)?

3. The results in Figure 6 are interesting. However, I only think panels C and D need to be shown as A and B are the same data. In addition, the authors should include more information for this analysis. In particular, what are the genetic differences between RRV G1-4? Were stocks of RRV generated on Vero cells like the BFV isolates? One could imagine that if RRV was passaged on insect cells, it may have an advantage on the C6/36 cells while the Vero derived BFV does not.

Along these lines, one issue with these results may be that the G2 (KO376-1) and G3 (SW10591) BFV are both viruses that have deletions in the 3’UTR (Table 3). It is known that the 3’UTR plays important roles in alphavirus replication in insect cells (site papers) and could be influencing your results. The authors should complete the same kinetic and plaque size experiments with a G2 and G3 isolate that has a complete 3’UTR. This will strengthen your results and add a level of function to the deletions if the phenotype changes.

4. I think the recombination analysis section needs a bit more to it. Why did you do this? Evidence in nature? If the authors prefer not to expand, as it is negative data, I suggest removing this section all together and adding a line or two to the discussion.

Author Response

Response to Reviewer 1's comments

Point 1:  Section 2.1 – I think this section would fit better in the introduction as it provides valuable facts about Australia and BFV and does not provide a "method"

Response 1: The authors agree that the information provided in Methods Section 2.1 is better suited for the introduction. Information from Section 2.1 has now been incorporated in the introduction.

Point 2:  I know it is referenced, but the authors should include details about RRV in the materials and methods. Are these viruses from an infectious clone, an isolate, how were they generated (passaged on Veros as well)?

Response 2: Clarification as to the passage history of the BFV and RRV isolates used in this investigation has been provided in the methods section. Isolates used in this study have a very similar passage history and were each passaged a single time on Vero cells prior to their use in experiments presented in this paper.

Point 3: The results in Figure 6 are interesting. However, I only think panels C and D need to be shown as A and B are the same data. In addition, the authors should include more information for this analysis. In particular, what are the genetic differences between RRV G1-4? Were stocks of RRV generated on Vero cells like the BFV isolates? One could imagine that if RRV was passaged on insect cells, it may have an advantage on the C6/36 cells while the Vero derived BFV does not.

Along these lines, one issue with these results may be that the G2 (KO376-1) and G3 (SW10591) BFV are both viruses that have deletions in the 3’UTR (Table 3). It is known that the 3’UTR plays important roles in alphavirus replication in insect cells (site papers) and could be influencing your results. The authors should complete the same kinetic and plaque size experiments with a G2 and G3 isolate that has a complete 3’UTR. This will strengthen your results and add a level of function to the deletions if the phenotype changes.

Response 3: The authors agree that information presented in plots A and B of Figure 6 is redundant, and these plots have been removed. The passage history of the BFV and RRV genotype representatives used in this study were very similar, involving minimal passage in both the C6/36 and Vero (or in one case a separate mammalian cell line, BHK cells) during isolation from mosquito homogenate. This has been clarified in the materials and in the discussion. A brief note on the genetic variation between RRV G1-4 has been provided in the methods section and is detailed more thoroughly in the cited article. The reviewers’ point about the influence of 3’UTR insertion and deletions on the results of the BFV replication kinetics studies is interesting and has now been mentioned in the discussion. As there are no wild-type G3 isolates available that lack the two large 3’UTR nucleotide insertions, investigating the impact on these particular insertions on G2 and G3 replications kinetics will need to be pursued in future experiments.

Point 4:  I think the recombination analysis section needs a bit more to it. Why did you do this? Evidence in nature? If the authors prefer not to expand, as it is negative data, I suggest removing this section all together and adding a line or two to the discussion.

Response 4: The recombination analysis was conducted, and the results mentioned, to assure the reader that the phylogenetic relationships defined in the paper were not confounded by the inclusion of known recombinants. The recombination results section has been expanded to clarify this. Recombination in alphaviruses has been observed previously:  historically, giving rise to the Western Equine Encephalitis virus group, and has been recently observed in Sindbis virus.

Reviewer 2 Report

In the manuscript entitled "Phylogenetic and timescale analysis of Barmah Forest virus as inferred from genome sequence analysis" by Michie et al., the authors present thirty novel viral genome sequences. The authors perform a comprehensive phylogenetic analysis utilizing Maximum-likelihood and Bayesian inference methods to reconstruct the phylogeny and find the time to the most recent common ancestor. Overall the article is well written, and my recommendation is to be accepted after a major review.

My key concern on this manuscript is the authors' definition of genotype, solely relying on a midpoint rooted tree where two "clades" consist of two or fewer sequences. Undeniably, the G1 sequence from PNG may be a new genotype, though further studies with more sequences isolated from that region would provide more insights in that matter. The presence of G2 bugs me as this could be an artifact of the midpoint rooting, and only more sequences could clear the relationship between "G2" and "G3". In order for them to be considered two distinct genotypes, I would expect more than just the clade distinction but an analysis of genetic distance, along with perhaps the presence of amino acid signatures that are well defined for these clades. I suggest the authors read the work of McNaughton et al. (2020) (Analysis of Genomic-Length HBV Sequences to Determine Genotype and Subgenotype Reference Sequences) and assess if there would be a way of better defining these genotypes. Otherwise, I recommend that these just be discussed as clades and the concept of genotype to be ignored.

I would expect the authors to have partitioned the dataset given that the sequences utilized are whole-genome sequences. The authors should have performed model tests for each individual gene rather than a single partition. Its been proven that unpartitioned datasets can result in spurious alignments, which will interfere with the phylogenetic analysis. I also advise the authors to be careful with the use of "bootstrap support," as bootstrap is not a measure of support but stability. There are ways to measure support, such as SH-aLRT, which can be found on IQ-TREE.

The authors discuss the presence of highly divergent UTRs found in the BFV sequences when compared to RRV. It would be great if they could shed some light on the presence of secondary structure at RNA level.

The "supplemental tree", figure 3, have several polytomies, with several partial and whole E2 genes, I advise it to be kept as a supplemental figure as it does not bring anything new. I also feel that the whole comparison with RRV falls out of the article's scope in the way it is written.

As a minor thing, QLD was never introduced as Queensland (row 300), I have assumed that that is what it means and have not checked for other acronyms.

Author Response

Response to Reviewer 2's comments:

Point 1: My key concern on this manuscript is the authors' definition of genotype, solely relying on a midpoint rooted tree where two "clades" consist of two or fewer sequences. Undeniably, the G1 sequence from PNG may be a new genotype, though further studies with more sequences isolated from that region would provide more insights in that matter. The presence of G2 bugs me as this could be an artifact of the midpoint rooting, and only more sequences could clear the relationship between "G2" and "G3". In order for them to be considered two distinct genotypes, I would expect more than just the clade distinction but an analysis of genetic distance, along with perhaps the presence of amino acid signatures that are well defined for these clades. I suggest the authors read the work of McNaughton et al. (2020) (Analysis of Genomic-Length HBV Sequences to Determine Genotype and Subgenotype Reference Sequences) and assess if there would be a way of better defining these genotypes. Otherwise, I recommend that these just be discussed as clades and the concept of genotype to be ignored.

Response 1: In an (unpresented) unrooted, radial phylogeny the three genetic groups, or ‘genotypes’, defined in this study were distinct. In terms of nucleotide divergence, the three genetic groups vary from between 1.8 – 3.2% nucleotide identity. For BFV, a mosquito-borne alphavirus that evolves under weak negative selection pressure over an estimated time-course of 80-years in a restricted geographical space, the authors argue that this seemingly minimal sequence divergence is sufficient to define BFV genotypes. The data presented in this paper is the largest collection of whole genomes currently available for BFV and sampled over a regular timeframe since its first identification in 1974. The nomenclature can be updated as new sequence data is made available in the future, possibly from other geographical regions including Papua New Guinea. Unique amino acid substitutions for each genotype is presented in Supplementary Table 2, which of course is currently limited by the representation of one and two genome sequences of G1 and G2, respectively.

Point 2: I would expect the authors to have partitioned the dataset given that the sequences utilized are whole-genome sequences. The authors should have performed model tests for each individual gene rather than a single partition. Its been proven that unpartitioned datasets can result in spurious alignments, which will interfere with the phylogenetic analysis. I also advise the authors to be careful with the use of "bootstrap support," as bootstrap is not a measure of support but stability. There are ways to measure support, such as SH-aLRT, which can be found on IQ-TREE.

Response 2: Partitioning the whole genome dataset as individual genes, by codon, revealed that the GTR model was most suitable in a RAxML analysis. Model testing of each individual gene dataset also verified that the GTR+G+I model was best suited for each individual gene region alignment (with the exception of the 6K and E3 gene, small uninformative gene regions, that had the GTR+G as the most ideal model). The GTR+G+I model was applied in the reconstruction of the whole genome sequence maximum likelihood phylogeny. The use of ‘support’ in describing high bootstrap values above nodes has been replaced with more appropriate language, and we thank the reviewer for the correction.

Point 3: The authors discuss the presence of highly divergent UTRs found in the BFV sequences when compared to RRV. It would be great if they could shed some light on the presence of secondary structure at RNA level.

Response 3: Further analysis of the secondary structures of the 3’UTRs revealed that the 58-nt insertion present in most ‘G3’ isolates composed a unique, large hairpin loop structure. A brief description of this has been added to the amended manuscript. The authors will conduct a more thorough analysis of secondary tertiary structures upon the complete sequencing of the 3”UTRs of the collection.

Point 4: The "supplemental tree", figure 3, have several polytomies, with several partial and whole E2 genes, I advise it to be kept as a supplemental figure as it does not bring anything new. I also feel that the whole comparison with RRV falls out of the article's scope in the way it is written.

Response 4: Figure 3 has now been made a supplementary figure (Supplementary Figure 1), as it did not add any further to the results of the study.

Point 5: As a minor thing, QLD was never introduced as Queensland (row 300), I have assumed that that is what it means and have not checked for other acronyms.

Response 5: We thank the reviewer for highlighting this error. The meaning of the abbreviation ‘QLD’ as Queensland has been added to the amended manuscript.

Reviewer 3 Report

Overview

The authors provide a much-needed investigation into the molecular epidemiology of Barmah Forest virus. The study is well described and the conclusions are supported by the results.

Minor comments

Throughout: there should be a space between the numerical value and unit symbol.

Line 16: characterise

Line 350: Harvey (WA)

Line 377: replace ‘experienced’ with ‘contained’ (or something similar)

Figure 6: The data from panels A and B are replicated in panels C and D and are therefore unnecessary. All of the relevant comparisons are contained in panels C and D. As such, panels A and B should be deleted.

Line 502: of plaques

Figure 8: the data presented in Figure 8 is the same as Figure 7. The significance values/lines should be transferred to Figure 7 so that all of this information can be presented in one figure.

Figure 9: This figure should be included as supplementary material. The inclusion of these cell culture photos does not add anything to the paper.

Author Response

Response to Reviewer 3's comments

Point 1: Throughout: there should be a space between the numerical value and unit symbol.

Response 1: This has been amended throughout the manuscript.

Point 2: Line 16: characterise, Line 350: Harvey (WA), Line 377: replace ‘experienced’ with ‘contained’ (or something similar), Line 502: of plaques

Response 2: All typos and word choice errors have been corrected, and we thank the reviewer for highlighting them.

Point 3: Figure 6: The data from panels A and B are replicated in panels C and D and are therefore unnecessary. All of the relevant comparisons are contained in panels C and D. As such, panels A and B should be deleted.

Response 3: Panels A and B have been removed from this figure. The figure now presents the graphs originally presented as panels C and D, showing the differences in replication kinetics of RRV and BFV isolates in C6/36 and Vero cells during the infection experiment.

Point 4: Figure 8: the data presented in Figure 8 is the same as Figure 7. The significance values/lines should be transferred to Figure 7 so that all of this information can be presented in one figure.

Response 4: The information presented in Figure 7 and Figure 8 have been kept separate, to more clearly present the information they are intended to convey. Figure 8 has now been made a supplementary figure, as to tidy up this section of the manuscript. 

Point 5: Figure 9: This figure should be included as supplementary material. The inclusion of these cell culture photos does not add anything to the paper.

Response 5: Figure 9 is now a supplementary figure, as it did not contribute anything to the results of the analysis, as the reviewer highlights in this comment.

Round 2

Reviewer 1 Report

I thank the authors for addressing the majority of my concerns. However, perhaps I missed it but the recombination section is not expanded as the authors say. The authors should expand this section as they mention in the responses. 

Author Response

Reviewer 2 Comment:

I thank the authors for addressing the majority of my concerns. However, perhaps I missed it but the recombination section is not expanded as the authors say. The authors should expand this section as they mention in the responses.

Author Response:

We thank the reviewer for highlighting this omission and apologise for the error. The recombination results section has now been expanded to clarify why this analysis was conducted.